# Skilled birth care uptake among women from socially disadvantaged minorities in the Kambata-Tambaro Zone, Southern Ethiopia

**Abebe Alemu** [1]*, **Biruk Assefa**[1], **Ritbano Ahmed**[1], **Hassen Mosa**[1], **Negesso Gebeyehu**[2]

**1** Department of Midwifery, College of Medicine and Health Sciences, Wachemo University, Hossana, Ethiopia, **2** Department of Midwifery, College of Medicine and Health Sciences, Mada-Walabu University, Robe, Ethiopia

* aalemu72@yahoo.com

**Data Availability Statement:** All data available in the document.

## Abstract

Globally in 2019, it was reported that 295,000 women die during pregnancy and childbirth every year. In Ethiopia, skilled birth care service uptake was low. Thus, the study aimed to assess the magnitude of skilled birth care uptake, and associated factors among women from socially disadvantaged minorities in the Kambeta-Temabaro Zone, Southern Ethiopia. A multistage sampling procedure was employed to enroll 521 study participants. Data were entered using EPI-INFO and SPSS-21 for analysis. Bivariate and multivariate analysis was done and the degree of association was assessed using odds ratios with a 95% confidence interval and variables with p values <0.05 were declared statistically significant. The magnitude of skilled birth care service uptake among women from socially disadvantaged minorities was 19%. Maternal education, occupation, awareness of birth care, pregnancy plan, number of births, mothers' lifestyle, and social subordination were significantly associated with skilled birth care service uptake in the study area. Thus, awareness creation on skilled birth, improving access to education for women, increasing the employability of women, and conducting community forums to avoid social discrimination against minorities are highly recommended.

## Background

Recent estimates of maternal mortality ratios (MMRs) suggest a substantial decline in recent years in low and middle-income countries, but it is far off the target [1]. In 2019, it was reported that 295,000 women die during pregnancy and childbirth every year globally [2]. Poor maternal health care service uptake remains a significant problem in low and middle-income countries [3].

A skilled birth care service is a key element of the safe motherhood service that aims to improve maternal and newborn wellbeing. World Health Organization recommended that providing focused antenatal care services during pregnancy can improve maternal and newborn wellbeing [4]. Analysis of a national survey of seven count-down countries shows that

**Funding:** Wachemo University funded the research (30,000 to AA). The funder has no role in study design, data collection and analysis, decision to publish, or preparation of the manuscript.

**Competing interests:** The authors have declared that no competing interests exist.

antenatal care is critical for improving maternal and newborn health [5]. In developing countries, maternal health care utilization varies due to different factors, with most findings showing differences between affluent and poor women and between women living in urban and rural areas [6].

In sub-Saharan African countries, the multi-country analysis revealed that social inequalities in maternal health care services lead to insufficient progress on maternal mortality and morbidity [7]. A study conducted in Ghana on accessibility and utilization of skilled birth care services showed that large gradients of inequities exist between geographic regions, urban and rural areas, and different socio-demographic, religious, and ethnic groups [8].

According to the analysis of the Ethiopian Demographic Health Survey 2016, on maternal health service utilization, 11.7% used skilled delivery care and 9.7% of women had postnatal care services. Education of women, household wealth, women's autonomy, and residence had a significant association with the uptake of skilled maternity care [9]. Studies have revealed that socioeconomic and cultural factors, such as women's age, ethnicity, education, culture, need for care, and decision-making power, largely account for variations in maternal health care utilization [10].

A study conducted previously revealed that the parity, literacy status of women, average monthly family income, media exposure, decision, where to give birth, perception of distance to health institutions, and antenatal care visits were associated with skilled birth service utilization [11]. Skilled birth care utilization inequities persist among vulnerable minorities because the services are not socially and culturally sensitive; even though, health policies proclaim that, every woman, everywhere has the right to have good quality care before, during pregnancy, and childbirth [12, 13].

Unless the health care service takes into account the necessary beliefs, attitudes, or culture of all pregnant women, even the best and most physically accessible may remain underused [14]. Social discrimination in the health care system directly contributes to the process of marginalization by perpetuating negative stereotypes and social isolation [15]. Some studies revealed that disadvantaged women are vulnerable to poor health care and negative stereotyping of poverty, social status, parenting styles, preferences, and unsupportive feedback from caregivers. Disadvantaged women consistently report constraints in access to skilled birth care that range from physical and psycho-social barriers to economic constraints [16–19].

According to the Ethiopian Demographic Health Survey 2016, the maternal mortality rate remains high at 412 per 100,000 live births. On top of that, there is a discrepancy in skilled birth care utilization among the highest wealth and lowest wealth social groups (70% vs. 11%) and urban and rural areas in the county (80% vs. 21%) [20]. Addressing contributing factors of social marginalization could play an important role to improve the uptake of skilled birth care services among minorities in the study area. Thus, this study aimed to assess the magnitude of skilled birth care uptake and associated factors among women from socially disadvantaged minorities in the Kambata-Temabaro zone, Southern Ethiopia.

## Methods and materials

### Study area, design, period, and population

The community-based cross-sectional study design was conducted in the Kambata-Tambaro zone, Southern Nations, Nationalities, and People Region, Ethiopia from March 01 to April 30, 2019. All postnatal women in socially disadvantaged minorities in the study area were considered as the source population and also, selected post-natal women were taken as the study population.

## Sample size determination

A single population formula was used to determine the sample size. The computation was made with the inputs of a 95% confidence level $(Z\alpha/2 = 1.96)$ [21], a margin of error $(d = 5\%)$, the prevalence of skilled care $(P = 29\%)$ [22], and the design effect (DE) of 1.5. Finally, a 10% nonresponse rate was considered to determine the total sample size (N = 521).

## Sampling procedures

To enroll the study population, a multistage sampling technique was used and the sample size was proportionally allocated to the selected three rural districts in the Kambata-Tambaro Zone. From each selected district, six (06) kebeles' were selected randomly. The list of postnatal women was found from registration logbooks in each kebele from Health Extension workers. Finally, study participants were enrolled using a simple random sampling method using randomly generated numbers from delivery registration logbooks in health posts.

## Inclusion and exclusion criteria

All women who gave birth within the last six weeks among socially disadvantaged minorities were included in the study population. However, those who were critically ill during data collection were excluded.

## Data collection procedures and tools

An interviewer-administered questionnaire was used to collect the data. The questionnaire comprised socio-demographic characteristics, and maternal health care utilization during prenatal, intrapartum, and postnatal periods. The questionnaire was adapted from Ethiopian Demographic Health Surveys, and other previous studies then after, it was translated into the local language by expertise's. A pilot study was conducted and necessary amendments were incorporated.

## Data management and analyses

The data entry was done using Epi-Info version 3.6 software and exported to Statistical Package for Social Science (SPSS)-21 for analysis. Descriptive statistics findings are presented in tables with frequencies and percentages. Both bivariate and multiple variable logistic regression analyses were applied to determine the association of independent variables with skilled birth care uptake. Those variables with p<0.25 at bivariate logistic regression were taken into multiple variable logistic regression model. The degree of association between independent and outcome variables was assessed using odds ratios with 95% confidence intervals, and the s with a p-value <0.05 were declared statistically significant. The model fitness was checked using Pearson's Chi-square with a value of 3.45 and a significance of 0.026.

## Operational definition

**Social marginalization.**   Is defined as the social disadvantage/subordination of an individual or group being disadvantaged or subordinated to access the skilled birth care services in the community [17, 19].

**Skilled birth.**   Refers to the care provided to a woman and newborn during childbirth by an accredited and competent health care provider.

**Minority groups.**   Describe groups that are subordinated or lack access to society due to their perceived identities, place of residence, friendship association, and daily activities.

**Anticipated stigma.** Women may avoid seeking birth care services, as they have anticipated they will be exposed to stigma due to their perceived status as coming from a minority group.

## Ethical considerations

Ethical approval for the research was received from the College of Medicine and Health Science, Wachemo University as referred to, ref. No: WCU/CMHS/17/2019. Written informed consent was obtained from study participants to collect data and the information attained was kept anonymous, thereby, ensuring confidentiality.

## Result

Of the 521 study participants, five hundred ten (510) responded to the questionnaire completely, resulting in a response rate of 97.8% [CI: 3.2–9.5]. The mean age of the respondents was 28.6 (+SD 4.8) years. Over 88 percent (88.8%) of the respondents were unable to read and write, and 95% were not employed [Table 1].

### Skilled birth care service utilization

The magnitude of skilled birth care uptake among women in disadvantaged minorities (19%) and four hundred fifteen (81%) of women didn't uptake skilled birth care in health facilities. Three hundred thirty (64.7%) of respondents didn't have awareness of skilled birth care. Among respondents who used skilled birth care, forty-five (47.4%) claimed that the service

**Table 1. Respondents' socio-demographic characteristics in the Kambata-Temabaro Zone, Southern Ethiopia, 2019 [n = 510].**

| Variables | Categories | Frequency | Percentage (%) |
|---|---|---|---|
| Respondents' age (mean = 28.6 +4.8) | <19 years | 106 | 20.78 |
| | | | |
| | 20–34 years | 308 | 60.39 |
| | >35 | 96 | 18.82 |
| Respondents religion | Protestant | 311 | 60.98 |
| | Muslim | 50 | 9.8 |
| | Catholic | 16 | 3.13 |
| | Orthodox | 13 | 2.54 |
| | No religion | 120 | 23.52 |
| Respondents' educational level | Can't read and write | 453 | 88.8 |
| | Primary and secondary education | 50 | 9.8 |
| | Diploma and above | 7 | 1.37 |
| Respondents' occupation | Housewife | 485 | 95.09 |
| | Employed | 25 | 4.9 |
| Husbands occupation | Farmer | 439 | 86.07 |
| | Merchant | 53 | 10.39 |
| | Employed | 18 | 3.5 |
| Average monthly income | < 1000 | 326 | 63.92 |
| | 1001–2000 | 123 | 24.1 |
| | 2001–3000 | 40 | 7.8 |
| | >3000 | 21 | 4.1 |
| Family size | ≤ 5 | 198 | 38.8 |
| | > 5 | 312 | 61.2 |

**Table 2. Respondents' skilled birth care service utilization in the Kambata-Temabaro Zone, Southern Ethiopia, 2019 [n = 510].**

| Variable | Categories | Frequency | Percentage (%) |
|---|---|---|---|
| Heard about skilled birth care | Yes | 180 | 35.3 |
| | No | 330 | 64.7 |
| Number of birth or parity | 1–4 | 210 | 41.2 |
| | >5 | 300 | 58.8 |
| Skilled birth care service utilization | Yes | 95 | 19 |
| | No | 415 | 81 |
| The reason for home birth | Personal factor | 115 | 27.7 |
| | Social factors | 149 | 35.9 |
| | Health system factor | 151 | 36.38 |
| Postnatal care services used | Yes | 76 | 15 |
| | No | 434 | 85 |
| Maternal and or neonatal complication during last birth at home delivery | Yes | 100 | 24.1 |
| | No | 315 | 75.9 |
| Maternal and or neonatal complication during last birth at facility delivery | Yes | 4 | 4.2 |
| | No | 91 | 95.8 |
| The birth service was respectful | Yes | 50 | 52.6 |
| | No | 45 | 47.4 |
| Subordination in skilled birth care services | Yes | 52 | 54.7 |
| | No | 43 | 45.3 |
| Subordination hinder birth care service | Yes | 198 | 38.8 |
| | No | 312 | 61.2 |

was not respectful, and forty-three (45.35) stated that there was discrimination during skilled birth care service in the facilities. Concerning social subordination during health care service uptake, 198(38.8%) of the participants claimed that disrespect or discrimination during birth service is the major problem of skilled birth care utilization in health facilities [Table 2].

## Factors associated with skilled birth care uptake

Multivariable logistic regression analysis showed that skilled birth care service utilization among women from socially disadvantaged minorities was significantly associated with maternal education, occupation, and awareness of skilled birth care, pregnancy plan, number of births, mothers' lifestyle, and social discrimination [Table 3].

Mothers' educational status showed statistically significant association with skilled birth care utilization. Mothers' who had a diploma or above had odds 3.5 times higher to utilize the birth care service than mothers who could not read and write [adjusted OR = 3.5; 95% CI: 1.79–2.03].

Respondent's employment status was significant association with skilled birth care uptake among women from disadvantaged minorities in the study area. Mothers" who were employed had odds 2.5 times higher to have skilled birth care services from health facilities compared to non-employed mothers [adjusted OR = 2.5; 95% CI: 1.50–6.51].

Planned pregnancy was found to have a significant association with the utilization of skilled birth care services. Mothers' who had unplanned pregnancies 57% reduced odds to utilize skilled birth care services in health facilities than their counterparts [adjusted OR = 0.43; 95% CI: 2.29–3.89]. Respondent's awareness on skilled birth care had a significant association with service uptake. Mothers' who had awareness of skilled birth care services had odds 4.4 times higher of utilizing skilled birth care services in health facilities than those who did not have

**Table 3. Factors associated with skilled birth care uptake among women in socially disadvantaged minorities, Kambata-Tambaro zone, Southern Ethiopia, 2019.**

| | | Skilled birth care | | | |
|---|---|---|---|---|---|
| | | | Used | | |
| Variables | | Yes | No | COR | aOR |
| **Education level of mother** | | | | | |
| Can't read and write | | 253(55.8) | 200(44.2) | 1 | 1 |
| Primary& Secondary | | 30(62.5) | 20(37.5) | 1.8(2.82–4.33)* | 1.2(2.2–5.02) |
| Diploma and above | | 4(57.1) | 3(42.9) | 2.4(1.02–2.03)* | 3.5(1.9–2.03) |
| **Occupation** | | | | | |
| Housewife | | 100(20.6) | 385(79.4) | 1 | 1 |
| Employed | | 20(80) | 5(20) | 1.28(1.40–2.09)* | 2.49(1.50–6.5) |
| **Pregnancy planned** | | | | | |
| | Yes | 55(57.9) | 40(42.1) | 1 | 1 |
| | No | 105 (25.3) | 310(74.7) | 0.86 (1.35–2.39)* | 0.43(2.59–3.89) |
| **Antenatal care visit** | | | | | |
| | Yes | 65(56.5) | 50(43.5) | 1 | |
| | No | 115(29.1) | 280(78.9) | 0.321 (2.36–4.83)* | 0.75(1.6–3.03) |
| **Respondents life-style** | | | | | |
| | Yes | 118 (37.1) | 200(62.9) | 0.54(2.52–3.25)* | 0.46(2.4–3.16) |
| | No | 104(54.1) | 88(45.8) | 1 | 1 |
| **Awareness on skilled birth care** | | | | | |
| | Yes | 102(56.6) | 78(43.3) | 1.41(1.14–2.39)* | 4.4(4.2–7.02) |
| | No | 95(28.8) | 235(71.2) | 1 | 1 |
| **Number of birth** | | | | | |
| 1–4 | | 104(49.5) | 106(50.5) | 2.13(1.16–4.09)* | 3.23(2.03–5.65) |
| | > 5 | 186(62) | 114(38) | 1 | |
| **Respectful birth care** | | | | | |
| | Yes | 35(70) | 15(30) | 2.12(6.23–8.09)* | 5.05(2.36–6.89) |
| | No | 33(73.3) | 12(26.7) | 1 | 1 |
| **Does social subordination affect service uptake** | | | | | |
| | Yes | 80(40.4) | 118(59.6) | 0.35(6.23–8.09)* | 0.23(3.06–4.70) |
| | No | 150(48.1) | 162(51.9) | 1 | 1 |

* = p≤ 0.25 COR = Crude Odds ratio

** = p≤ 0.05 aOR = Adjusted odd ratio

awareness [adjusted OR = 4.4; 95% CI: 4.02–7.02]. The number of birth or parity of respondents had a significant association with the uptake of skilled birth care. Mothers who had one to four births had odds 3.2 times higher to use skilled birth care services in health facilities as compared with those who had five births and above [adjusted OR = 3.23; 95% CI: 2.03–5.65].

The life style of the mothers in disadvantaged minorities was significantly associated with skilled birth care uptake. Mothers' who had culturally unique lifestyle had 54% reduced odds to uptake skilled birth care services than their counterparts [adjusted OR = 0.46; 95% CI: 2.47–3.16].

Social subordination of mothers' from disadvantaged minorities was significantly associated with skilled birth care uptake in the study area. Respondents' who were socially subordinated had 77% reduced to use skilled birth care service utilization than mothers who didn't claim as socially subordinated [adjusted OR = 0.23; 95% CI: 3.06–4.70].

## Discussion

The government of the Federal Republic of Ethiopia has implemented a health policy that provides free maternal health care services for all women during pregnancy, during labor, and postnatal periods in governmental health facilities. The government has been implementing health policies that state 'health care for all, but still there were disadvantaged minorities for equitable health care service utilization. Hereby, this study revealed the result of a community-based cross-sectional study that aimed to determine the magnitude and associated factors with skilled birth care uptake among women from socially disadvantaged minorities in the Kambata_Tambaro Zone, Southern Ethiopia.

In this study, the magnitude of skilled birth care utilization among women from socially disadvantaged minorities in health facilities was 19%. The finding of this study was found to be lower when compared to previous studies conducted in Ethiopia(48%) and Timor-Leste (25%) [23, 24]. However, the magnitude is higher as compared to the study done in Northwest Ethiopia (13.8%) [25]. This difference might be due to differences in the study period, approach, demographic characteristics, and social status of study participants, and also it offers insight on whether health care services interventions are not yet effective for socially disadvantaged minorities in the study area.

In this study, mothers' educational status and occupation were found to be associated factors for skilled birth care service utilization among study participants from health facilities. This finding was similar to other previous study findings in Gedeo Zone, and Tigray region, Ethiopia respectively [26, 27]. These similarities of predictors of skilled birth care uptake indicate that women's access to education was significant for efficient health care services utilization. Consequently, accessing opportunities of education and employability for women may improve the uptake of skilled birth care services in the community, and also it might break the cultural taboo that hinders health care service uptake.

In this study, findings showed that the awareness of women towards skilled birth care, and the number of births were found to be associated factors for care utilization of skilled birth among the disadvantaged mothers in the study area. These findings are sustained with previous studies results conducted in Holata town and Northwest Ethiopia respectively [11, 25]. These studies' findings correspondence might be due to mothers' awareness on skilled birth care utilization across the different communities. Therefore, health education and promotion for mothers on skilled birth care are crucial to improving the uptake of birth care services among mothers from disadvantaged minorities.

In this study, social subordination within society was found to be a negatively affecting factor in the uptake of birth care services among women from disadvantaged minorities in the study area. The finding of the study was consistent with previously conducted research findings in different places and time [10, 17, 18].

The resemblance of the findings indicated that social subordination has a comparable negative effect on the uptake of skilled birth care services in different communities, and women within the disadvantaged minorities were more disadvantaged in the service. Even though, there were health policies that state health for all; conversely, mothers from subordinated minorities have been disadvantaged by this service due to distorted self-esteem, fear of stigma, and subordination by health care providers and communities.

## Conclusion

The prevalence of skilled birth care service utilization among disadvantaged mothers was found to be low [19%] in the Kambata-Tambaro zone, southern Ethiopia. Mothers' education, occupation, awareness of skilled birth care, number of births, and social subordination were significantly associated with skilled birth care service uptake in health facilities. Thus, awareness creation on skilled birth, improving access to education for women, increasing employability of mothers and conducting community forums to avoid social subordination against minorities are highly recommended.

## Acknowledgments

The authors extend their gratitude to study participant, data collectors, supervisors, and Wachemo University.

## Author Contributions

**Conceptualization:** Abebe Alemu, Biruk Assefa.

**Data curation:** Ritbano Ahmed.

**Formal analysis:** Abebe Alemu, Ritbano Ahmed.

**Methodology:** Abebe Alemu, Biruk Assefa, Negesso Gebeyehu.

**Software:** Abebe Alemu, Ritbano Ahmed, Hassen Mosa.

**Supervision:** Hassen Mosa.

**Validation:** Ritbano Ahmed.

**Writing – original draft:** Abebe Alemu.

**Writing – review & editing:** Abebe Alemu, Negesso Gebeyehu.

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
