## [Decision Letter · Decision Letter 0]

13 Jul 2022

PGPH-D-22-00283

Skilled birthing care uptake among women from socially marginalized minorities in the Kambata-Tembaro Zone, Southern Ethiopia

Dear Dr. Alemu,

Thank you for submitting your manuscript to PLOS Global Public Health. This manuscript has been assessed by two external reviewers. After careful consideration, we feel that it has merit but does not fully meet PLOS Global Public Health’s publication criteria as it currently stands. Therefore, we invite you to submit a revised version of the manuscript that addresses the points raised during the review process.

We look forward to receiving your revised manuscript.

Kind regards,

Julia Robinson

Executive Editor

Journal Requirements:

1.  Please ensure that Funding Information matches with Financial Disclosure Statement.

2. Please provide separate figure files in .tif or .eps format and remove the embedded figures within the manuscript file.

Additional Editor Comments (if provided):

Reviewers' comments:

Reviewer's Responses to Questions

**Comments to the Author**

1. Does this manuscript meet PLOS Global Public Health’s publication criteria? Is the manuscript technically sound, and do the data support the conclusions? The manuscript must describe methodologically and ethically rigorous research with conclusions that are appropriately drawn based on the data presented.

Reviewer #1: Partly

Reviewer #2: Partly

2. Has the statistical analysis been performed appropriately and rigorously?

Reviewer #1: No

Reviewer #2: No

3. Have the authors made all data underlying the findings in their manuscript fully available (please refer to the Data Availability Statement at the start of the manuscript PDF file)?

Reviewer #1: Yes

Reviewer #2: Yes

4. Is the manuscript presented in an intelligible fashion and written in standard English?

Reviewer #1: Yes

Reviewer #2: No

5. Review Comments to the Author

Reviewer #1: It is helpful to the reader of your paper if you can show the frequencies (percentages) by the outcome variable. This applies to Tables 1, 2, and 3.

I do not think that Figure 1 is useful.

In table 3, the total number of the education level of the mother is 540 (253+200+50+30+4+3) > 510. Please check the frequencies.

Did you considered a survey design techniques when you performed the statistical analysis?

Did you use standard logistic regression model or a logistic regression for a survey design?

On page 9, "Both bivariate and multiple variable logistic regression analyses were used to determine the association of independent variables with outcome variables." Please define the outcome variable.

Please define your primary aim?

Was the sample size determination performed for analysis in Table 3? Please clarify.

Do you have a reference for the sample size determination?

Reviewer #2: The subject matter of this manuscript is very critical for maternal and child health and accurate and adequate findings of such a study will add important knowledge to the field. The objective of the study was clear and there appears to be a lot of effort put into the study. There are some important findings from the study that are very useful. However, there are a number of issues with regards to the data analysis, the preparation of the manuscript, and the overall style of the manuscript. I have attached a copy of my review to this report for your kind review and necessary actions. From the methodology, ethical review, findings, and discussion sections, you need to make a major revision to improve the manuscript. With the current version of the manuscript, I cannot recommend it for publication until a major revision is completed. The whole manuscript requires professional editing to make it more understandable to the general reading public. I recommend that you download Grammarly and upload the manuscript into it so it can offer professional editing guidelines. Please, also look for examples of articles published by Plos Global Public Health for guidance.

6. PLOS authors have the option to publish the peer review history of their article (what does this mean?). If published, this will include your full peer review and any attached files.

**Do you want your identity to be public for this peer review?** For information about this choice, including consent withdrawal, please see our Privacy Policy.

Reviewer #1: No

Reviewer #2: No

---

## [Decision Letter · Decision Letter 1]

10 Oct 2022

Skilled birth care uptake among women from socially disadvantaged minorities in the Kambata-Tambaro Zone, Southern Ethiopia

PGPH-D-22-00283R1

Dear Dr Alemu,

We are pleased to inform you that your manuscript 'Skilled birth care uptake among women from socially disadvantaged minorities in the Kambata-Tambaro Zone, Southern Ethiopia' has been provisionally accepted for publication in PLOS Global Public Health.

Best regards,

Julia Robinson

Executive Editor

Reviewer Comments (if any, and for reference):

Reviewer's Responses to Questions

**Comments to the Author**

1. If the authors have adequately addressed your comments raised in a previous round of review and you feel that this manuscript is now acceptable for publication, you may indicate that here to bypass the “Comments to the Author” section, enter your conflict of interest statement in the “Confidential to Editor” section, and submit your "Accept" recommendation.

Reviewer #2: All comments have been addressed

2. Does this manuscript meet PLOS Global Public Health’s publication criteria? Is the manuscript technically sound, and do the data support the conclusions? The manuscript must describe methodologically and ethically rigorous research with conclusions that are appropriately drawn based on the data presented.

Reviewer #2: Yes

3. Has the statistical analysis been performed appropriately and rigorously?

Reviewer #2: Yes

4. Have the authors made all data underlying the findings in their manuscript fully available (please refer to the Data Availability Statement at the start of the manuscript PDF file)?

Reviewer #2: Yes

5. Is the manuscript presented in an intelligible fashion and written in standard English?

Reviewer #2: Yes

6. Review Comments to the Author

Reviewer #2: After reading through the entire manuscript several times, I have seen that the comments I made in previous review have been adequately addressed. Again, the subject matter of this research and manuscript is very important for public health professionals in the prevention of both maternal and infant mortality and improving maternal and child health. A little more editing will help greatly in perfecting the manuscript.

7. PLOS authors have the option to publish the peer review history of their article (what does this mean?). If published, this will include your full peer review and any attached files.

**Do you want your identity to be public for this peer review?** For information about this choice, including consent withdrawal, please see our Privacy Policy.

Reviewer #2: No
